# Influence of Beam Power on Young's Modulus and Friction Coefficient of Ti–Ta Alloys Formed by Electron-Beam Surface Alloying

**Stefan Valkov** [1],*[iD]**, Dimitar Dechev** [1]**, Nikolay Ivanov** [1]**, Ruslan Bezdushnyi** [2]**, Maria Ormanova** [1]
**and Peter Petrov** [1]

[1]  Institute of Electronics, Bulgarian Academy of Sciences, 72 Tsarigradsko, Chaussee, 1784 Sofia, Bulgaria; dadechev@abv.bg (D.D.); n_ivanov@ie.bas.bg (N.I.); m.ormanova@ie.bas.bg (M.O.); pipetrov@ie.bas.bg (P.P.)
[2]  Faculty of Physics, St Kliment Ohridski University of Sofia, 1164 Sofia, Bulgaria; rb@phys.uni-sofia.bg
*  Correspondence: stsvalkov@gmail.com; Tel.: +359-2-979-5914

**Abstract:** In this study, we present the results of Young's modulus and coefficient of friction (COF) of Ti–Ta surface alloys formed by electron-beam surface alloying by a scanning electron beam. Ta films were deposited on the top of Ti substrates, and the specimens were then electron-beam surface alloyed, where the beam power was varied from 750 to 1750 W. The structure of the samples was characterized by scanning electron microscopy (SEM), energy-dispersive X-ray spectroscopy (EDX), and X-ray diffraction (XRD). Young's modulus was studied by a nanoindentation test. The coefficient of friction was studied by a micromechanical wear experiment. It was found that at 750 W, the Ta film remained undissolved on the top of the Ti, and no alloyed zone was observed. By an increase in the beam power to 1250 and 1750 W, a distinguished alloyed zone is formed, where it is much thicker in the case of 1750 W. The structure of the obtained surface alloys is in the form of double-phase α′and β. In both surface alloys formed by a beam power of 1250 and 1750 W, respectively, Young's modulus decreases about two times due to different reasons: in the case of alloying by 1250 W, the observed drop is attributed to the larger amount of the β phase, while at 1750 W is it due to the weaker binding forces between the atoms. The results obtained for the COF show that the formation of the Ti–Ta surface alloy on the top of Ti substrate leads to a decrease in the coefficient of friction, where the effect is more pronounced in the case of the formation of Ti–Ta surface alloys by a beam power of 1250 W.

**Keywords:** Ti–Ta alloys; electron-beam surface alloying; Young's modulus; coefficient of friction; biomedical implant material

## 1. Introduction

Titanium and its alloys are commonly used in the field of modern biomedicine and for the manufacturing of implants due to their biocompatibility, superior corrosion resistance, and static and fatigue strength [1–3]. However, some health problems related to the release of metallic ions exist, which could lead to adverse reactions and implant failure [4]. The existence of cracks and other structural features are critical for the understanding of failure mechanisms and could be evaluated by synchrotron radiation X-ray-computed microtomography [5,6]. Furthermore, Young's modulus of these materials significantly differs from that of the human bones, which can cause the resorption of adjacent bone tissues [7]. The ideal implant should exhibit low Young's modulus (very close to that of the human bones), excellent wear resistance, and biocompatibility [8]. The discussed requirements for the implant material depend mostly on the surface properties of the alloys, and therefore, can be achieved by an appropriate technique for surface modification.

Currently, the modification of the surface properties of the materials can be achieved by a number of methods, such as thin film deposition [9], ion implantation [10], treatment

and alloying of the surface by high energy fluxes, such as electron beam [11–13], laser beam [14,15]. The electron-beam surface alloying receives a lot of attention due to the possibility of precise control of the technological conditions, short process time, uniform distribution of the energy of the electron beam, and so on [16]. For this technology, the electrons interact with the surface of the modified material, where their kinetic energy is transfer into heat. The technological conditions can be optimized to melt the treated surface and to form a melt pool. The alloying elements are incorporated into the molten material, and after its solidification, a surface alloy with significantly improved functional properties is formed [16].

Nowadays, the Ti–Ta alloys are considered as very promising for a large number of biomedical applications due to their high biocompatibility, corrosion resistance, high specific strength, toughness, and fracture resistance. Furthermore, this material is known as a high-temperature shape memory alloy. The authors of [17] reported that the Ti–Ta alloys are characterized by low Young's modulus and better strength in comparison with the pure Ti, and the corrosion capacity of this material exceeds that of Ti6Al4V, which is known as a standard material for biomedical applications. In work [18], the Ti–Ta alloys were manufactured by selective laser melting, and the results showed that the Ta amount is of major importance for the phase composition, microstructure, mechanical, and corrosion properties. The Ti–25Ta exhibited superior mechanical properties and resistance to corrosion. According to the authors of [19], Young's moduli of Ti–Ta alloys are in the range from 115 to 65 GPa, where the measured values strongly depend on the Ta concentration [19]. The same authors [19] considered this alloy promising for implant manufacturing. Furthermore, the authors of [20] have studied the friction coefficient of Ti–Ta and Ti–Ta–Ru alloys, and their results showed that Ti–10Ta exhibited the lowest coefficient of friction (COF), and the concentration of the alloying elements has a significant effect on the COF.

The authors of [21] have successfully demonstrated the possibilities of the formation of Ti–Ta surface alloys on TiNi substrates by pulsed electron-beam melting of the TiTa films/TiNi substrate system. Their results showed that the elastoplastic properties were greatly improved. Furthermore, a sublayer with $Ti_3Ni_4$ particles, which play the role of physicomechanical properties stabilizer, and provided a transition between the substrate and the coating, was observed. Golkovski et al. [22] demonstrated the formation of Ti–Ta alloy on pure VT1-0 titanium substrate by an atmospheric electron-beam cladding of a mixture of Ti and Ta powders. Their results demonstrated that the rise in the Ta content leads to an improvement in the tensile strength. Further, the cladding of the Ti–Ta powders leads to an improvement in the corrosion properties in comparison with the uncladded substrate material.

Our previous study [23] was based on the formation of Ti–Ta surface alloys by electron-beam alloying of Ta films with Ti substrate, where the dimension of the scanning figure was varied. A smaller dimension of the scanning figure led to higher heat input, and the distribution of the Ta element within the Ti matrix was significantly more homogeneous. However, investigations of Young's modulus and coefficient of friction (COF) of the Ti–Ta surface alloys formed by electron-beam surface alloying of Ti substrate with Ta films are currently lacking in the scientific literature. As already mentioned, the elastic and friction properties are of major importance for the ideal implant material. Therefore, this study aims to investigate Young's modulus and coefficient of friction (COF) of Ti–Ta surface alloys formed by electron-beam surface alloying of Ti substrate with Ta film under different technological conditions. During the experiments, the electron-beam power was varied from 750 to 1750 W, and the results are discussed concerning the influence of the applied technological conditions (defined by the beam power) on the resultant structure of the obtained surface alloys and its effect on the elastic and friction properties.

## 2. Materials and Methods

Ti–Ta surface alloys were formed by selective electron-beam surface alloying, as on commercially pure α-Ti substrate with dimensions of 20 mm × 20 mm, and a thickness of 4 mm, Ta films with a thickness of 2.5 μm were deposited by direct current (DC) magnetron sputtering. During the deposition process, the discharge voltage was 440 V, the discharge current was 1 A, and the deposition time was 1 h. The samples were then selective electron-beam surface alloyed by a continuous electron beam, where a circular manner of scanning was used. Using this manner of scanning, the beam trajectory overlaps, leading to a significant increase in the lifetime of the melt pool and a more homogeneous distribution of the alloying elements within the alloyed zone. During the alloying process, the accelerating voltage was 50 kV; the electron beam current was 15, 25, and 35 mA, corresponding to a beam power of 750, 1250, and 1750 W, respectively; the speed of the sample motion was 5 mm/s, and the electron beam scanning frequency was 200 Hz. A scheme of the electron-beam surface alloying process is shown in Figure 1.

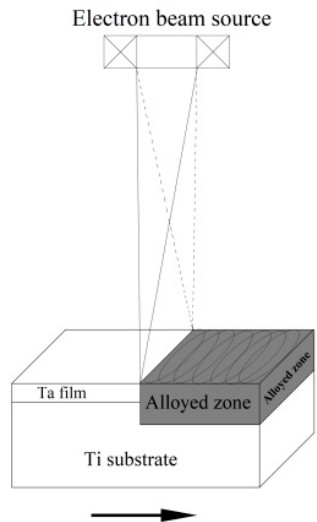

**Figure 1.** A scheme of electron-beam surface alloying of Ta film with Ti substrate.

The structure of the obtained specimens was studied by scanning electron microscopy (SEM, LYRA I XMU (Tescan), Brno, Czech Republic). The high voltage (HV) was 20 kV. During the measurements, back-scattered electrons were used. The chemical composition was investigated by energy-dispersive X-ray spectroscopy (EDX microanalyzer—Quantax 200, Bruker, Billerica, MA, USA). The EDX detector integrates a true standardless analysis with P/B ZAF quantitative corrections (Z = the atomic number correction factor; A = X-ray absorption correction factor; F = fluorescence correction factor).

The phase composition and crystallographic structure were studied by X-ray diffraction (XRD, URD 6 Seifert & Co diffractometer, GmbH, Ahrensburg, Germany), where Cu Kα radiation (1.54 Å) was used. The measurements were carried out within the range of 30–75° at 2 theta scale. The step was 0.1°, and the counting time was 10 s per step.

The Young's modulus of the obtained specimens was studied by nanoindentation, where a nanomechanical tester (Bruker, Billerica, MA, USA) was used. The number of indentations was 48 (4 lines with 12 indentations), where the spacing was 80 μm. The applied load was 200 mN.

The coefficient of friction was studied using a micromechanical wear test—ball-on-flat. The experiments were performed for 300 s with a load of 5 N at room temperature, where a hardened steel ball was used. The measurements were done according to ASTM D4518-91standard—standard test methods for measuring static friction of coating surfaces.

## 3. Results and Discussion

Figure 2a presents a cross-sectional SEM image of the electron-beam specimen processed with a beam power of 750 W. The chemical composition at different areas was studied by EDX; the results are shown in Figure 2b,c and are summarized in Table 1. It is obvious that the deposited Ta film remains unmelted, and no intermetallic surface structure in the system of Ti–Ta can be seen on the top of the specimen. This means that the technological condition used (i.e., a beam power of 750 W) is not capable of melting the surface of the specimen and dissolving the deposited Ta layer into the base Ti matrix. Therefore, the application of electron-beam surface alloying of Ta films with Ti substrate using a beam power of 750 W does not lead to the formation of a surface alloy in the system of Ti–Ta.

**Table 1.** Chemical composition of each measured area/point marked on the cross-sectional SEM images (Figures 2–4).

| Specimen | Area/Point | Ti, wt.% | Ta, wt.% |
|---|---|---|---|
| 750 W | Area1 | 1.9 ± 0.1 | 98.1 ± 2.9 |
| | Area 2 | 100.0 ± 2.6 | - |
| 1250 W | Area 3 | 80.9 ± 2.4 | 19.1 ± 0.8 |
| | Point 4 | 4.1 ± 0.2 | 95.9 ± 2.4 |
| 1750 W | Area 5 | 91.8 ± 2.7 | 8.2 ± 0.3 |

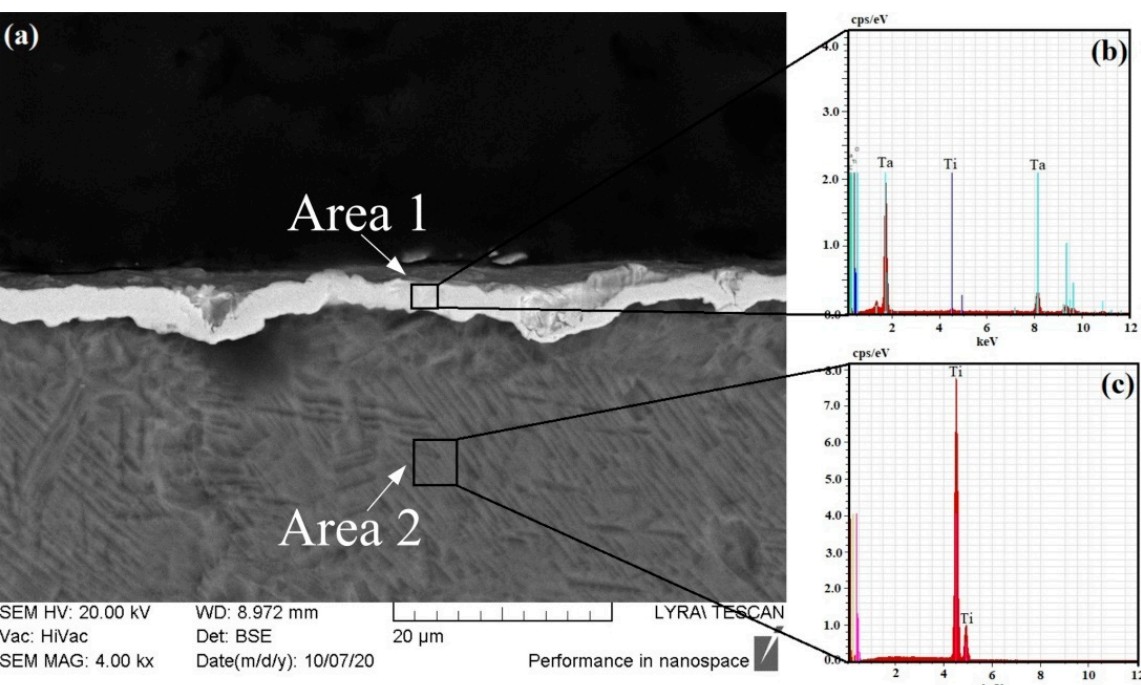

**Figure 2.** A cross-sectional SEM image of the sample processed with a beam power of 750 W (**a**) and corresponding EDX spectra taken from different regions (**b**,**c**).

A cross-sectional SEM image of the sample alloyed by a beam power of 1250 W is shown in Figure 3a. The chemical composition of the alloyed zone was studied by EDX, and the experimentally obtained spectra are shown in Figure 3b,c. The results obtained by EDX experiments are summarized in Table 1. The alloyed zone is marked as zone A, and its thickness is about 20 μm. The base Ti substrate is indicated as B. The results revealed that by using the above-mentioned technological conditions and a beam power of 1250 W, a distinguished alloyed zone in the system of Ti–Ta was formed. The increase in the beam power, from 750 to 1250 W, leads to an increase in the surface temperature of the processed

sample [24], melting of the Ta film, and the formation of a melt pool on the top of the specimen. After the solidification of the molten material, a surface alloy in the system of Ti–Ta is formed. However, some unmelted Ta particles exist within the alloyed zone (point 4 from Figure 3a and Table 1), meaning that the use of the above-mentioned electron-beam alloying technological conditions and a beam power of 1250 W does not lead to complete melting and dissolution of the Ta film into the Ti substrate. During the electron-beam surface alloying technology, an intense Marangoni convection exists, which is caused by the large high-temperature gradient in the melt pool, which is responsible for the melt homogenization [25]. However, in this case, the input energy density is low, leading to the very short lifetime of the melt pool and insufficient temperature gradient for complete homogenization. The results obtained by the energy-dispersive X-ray spectroscopy show that the formed alloyed zone is in the form of Ti–20 wt.% Ta.

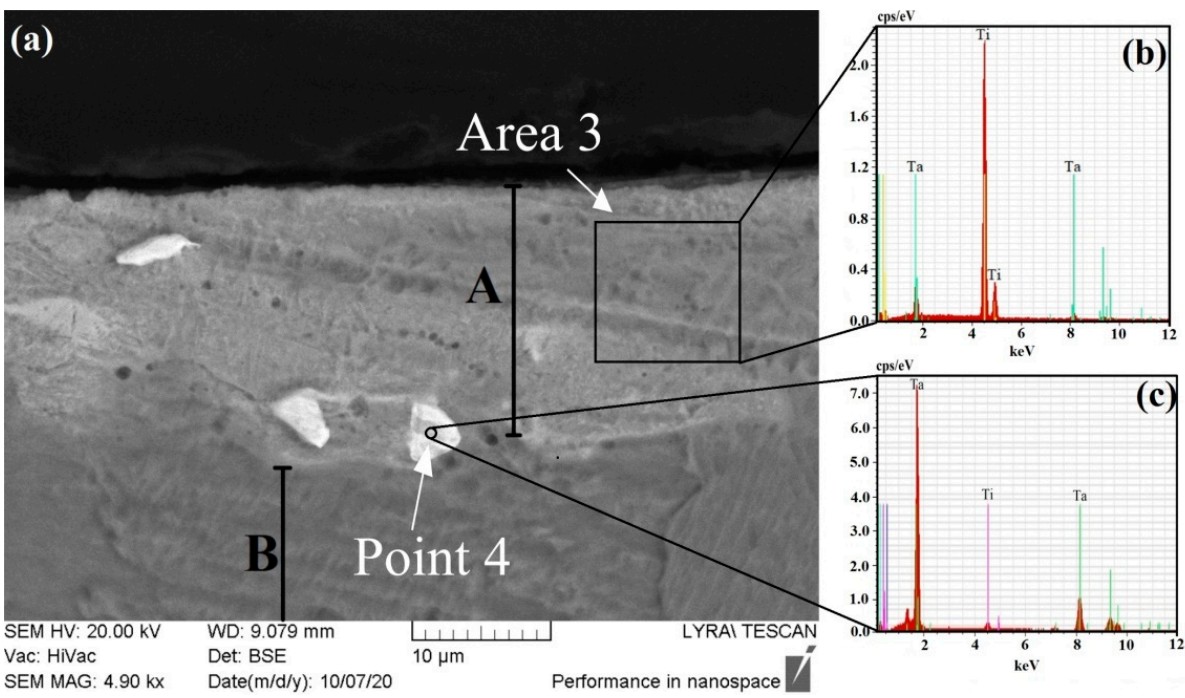

**Figure 3.** A cross-sectional SEM image of the sample processed with a beam power of 1250 W (**a**) and corresponding EDX spectra taken from different regions (**b**,**c**).

A cross-sectional SEM image of the sample alloyed by a beam power of 1750 W is shown in Figure 4a. The chemical composition of the alloyed zone is studied by EDX. The experimentally obtained spectrum is shown in Figure 4b, and the quantitative results are summarized in Table 1. The alloyed zone is indicated as zone A, while the base Ti substrate is marked as zone B. The thickness of the alloyed zone is about 90 μm and significantly exceeds that of the specimen alloyed by a beam power of 1250 W. No undissolved Ta particles can be seen, meaning that the use of the above-mentioned electron-beam surface alloying technological conditions and a beam power of 1750 W leads to a complete dissolution of the Ta film. The rise of the beam power leads to an increase in the surface temperature of the processed sample and a larger high-temperature gradient in the melt pool. In this case, the Marangoni convection is significantly intensified in comparison with the previous case (i.e., the alloying with a beam power of 1250 W), leading to the formation of a much more homogeneous structure without undissolved Ta particles. The results obtained by the EDX experiments show that the formed surface alloy is in the form of Ti–8 wt.% Ta. It is obvious that the amount of the Ta element is much lower in comparison with the case of the electron-beam surface alloying using a beam power of 1250 W. This could be attributed to the formation of a surface alloy with significantly higher thickness where the same amount

of alloying Ta element is distributed within a much larger alloyed zone. Furthermore, by using the higher beam power of 1750 W, the probability of evaporation of some amount of the alloying element during the alloying process is much higher, leading to a decrease in the concentration of this element.

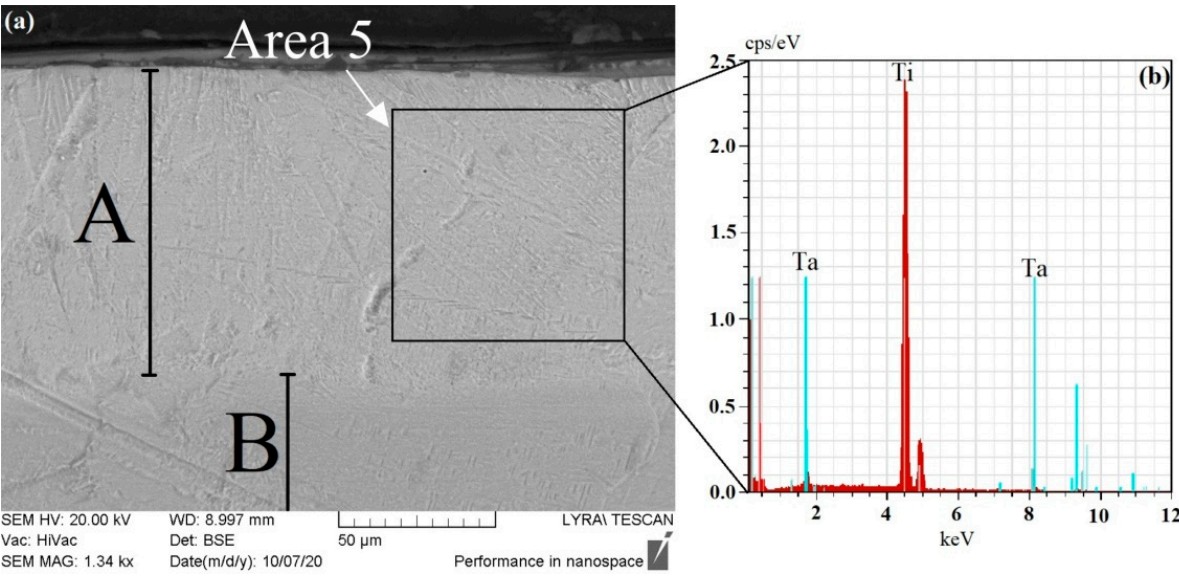

**Figure 4.** A cross-sectional SEM image of the sample processed with a beam power of 1750 W (**a**) and corresponding EDX spectrum taken from the alloyed zone (**b**).

The experimentally obtained X-ray diffraction patterns of the Ti–Ta surface alloys formed by electron beam power of 1250 and 1750 W are shown in Figure 5a,b, respectively. The analysis of the phase composition was done according to the ICDD (international center for diffraction data) crystallographic database. Both diffractograms are typical for polycrystalline materials. The analysis of the phase composition of the considered samples shows that in both cases, it is in the form of double-phase structure of α′ martensitic and β. The beta phase is characterized by a body-centered cubic (bcc) crystal structure and is known as a high-temperature modification of the titanium. Although the β phase is metastable, the addition of some amount of Ta, which is known as a beta stabilizing element, is responsible for the formation of this structure in a stable state at room temperature [26]. The α′ martensitic phase is characterized by a hexagonal close-packed (hcp) crystal structure and appears at α + β titanium alloys. The formation of non-equilibrium phases, such as the metastable martensitic α′, can be attributed to the very high cooling rate at the electron-beam alloying process [27]. It is known that the cooling rate can reach values of about $10^5$ K/s for the continuous mode [28]. These results are in agreement with those obtained by the authors of [18]. In Ref. [18], an investigation of the structure and properties of Ti-Ta alloys formed by selective laser melting was reported;it was concluded that the phase composition of the specimens is in the same form of α′ and β. Considering the peaks of the β, it is obvious that in the case of electron-beam alloying with a beam power of 1250 W, the diffraction maxima of the discussed phase are much stronger than that of the specimen obtained at a beam power of 1750 W. This means that the amount of the beta phase is larger in the case of electron beam surface alloying by a beam power of 1250 W than that of 1750 W. According to the authors of [18], an increase in the Ta content corresponds to a greater amount of the beta phase. It is known that the Ta element plays a role of a beta stabilizing element, and a higher concentration of the Ta corresponds to a larger amount of the beta phase. This is again in agreement with the results obtained by SEM/EDX and XRD experiments. As it was already discussed, the amount of Ta content is greater in the case of electron-beam surface alloying of the Ti substrate with Ta films by a

beam power of 1250 W than that of the 1750 W. Considering the diffraction pattern of the specimen alloyed with a beam power of 1250 W, beside the peaks of the discussed α′ and β phases (Ti–Ta structures), a diffraction maximum at about 33.2° at 2θ scale exists, which can be attributed to undissolved Ta. Tantalum exists in two crystalline forms, namely alpha and beta phases, and the identified Ta peak belongs to the beta structure. Although the beta phase is metastable and the bulk Ta exists in the form of almost entirely alpha phase, the authors of [29] found that the β phase can exist in a stable state in the form of thin films. This again correlates with our results showing a presence of undissolved β-Ta, which was deposited on the top of the Ti substrate as an alloying element and is consistent with SEM/EDX investigations.

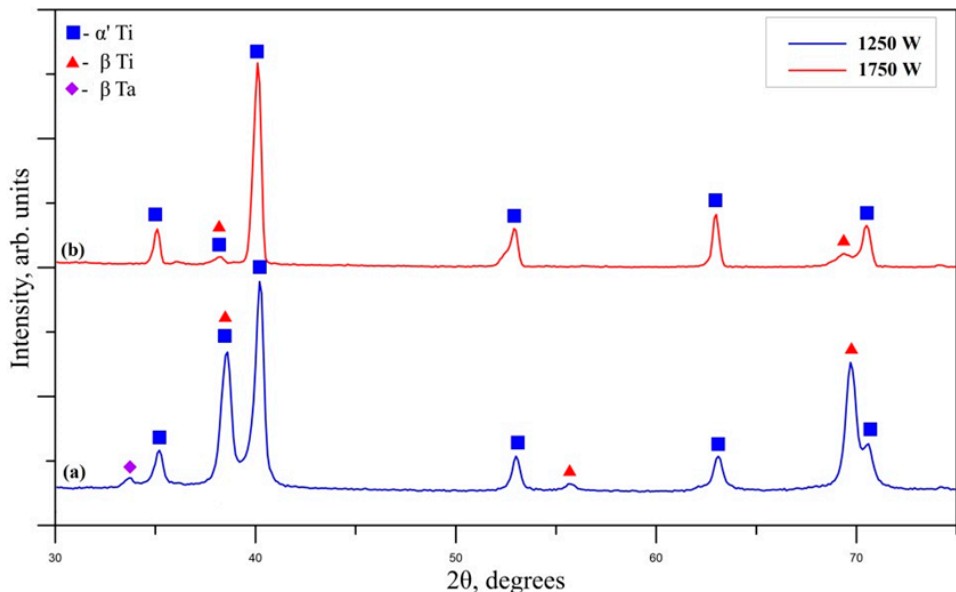

**Figure 5.** X-ray diffraction patterns of the specimens alloyed with a beam power of 1250 W (**a**) and 1750 W (**b**).

The Young's modulus of the samples alloyed by a beam power of 1250 and 1750 W, as well as the base Ti substrate, were studied by nanoindentation experiments. The results are presented in Figure 6 and are summarized in Table 2. The measured values of the Ti–Ta surface alloys are 60.3 ± 5.1 GPa in the case of alloying by a beam power of 1250 W and 59.1 ± 16.2 GPa at 1750 W. In both cases, Young's modulus is two times lower in comparison with that of the base Ti material (110 ± 9.8 GPa) and is much closer to that of human bones. According to the scientific literature, the chemical and phase composition have a serious influence on elastic properties [16,30–32]. It was reported that the beta phase is characterized by the lowest Young's modulus in comparison with the martensitic structures, as well as α-Ti in the titanium-based alloys. On the other hand, Young's modulus can be affected by the boding force among the atoms, which strongly depends not only on the crystal structure but also on the atomic distance. The bonding force between the atoms can be influenced by the alloying elements addition, heat treatment, etc. In the present study, the obtained surface alloys were formed by different technological conditions, i.e., alloying by a beam power of 1250 and 1750 W, meaning that the bonding forces should be different. For a better understanding of the bonding forces and atomic distances, unit cell volumes of hcp and bcc phases were experimentally evaluated from the XRD experiments, and the results are presented in Figure 7a,b. It can be seen from Figure 7a,b that the unit-cell volumes of the considered phases increase with an increase in the beam power, which, according to the authors of [33], is a normal feature for electron-beam processed materials. It can be concluded that the decrease in Young's modulus of the Ti–Ta specimens alloyed by a beam power of 1250 and 1750 W, respectively, is due to two different reasons. In the



case of electron-beam surface alloying of Ti substrate with Ta film by a beam power of 1250 W, the experimentally measured Young's modulus is about two times lower than that of the pure Ti material, where the observed decrease can be attributed to the formation of double-phase $\alpha'$ and $\beta$ structures, where the contribution of the $\beta$ phase is significant. As mentioned above, this phase is characterized by the lowest values of Young's modulus. At electron-beam surface alloying of Ti substrate/Ta film system by a beam power of 1750 W, the values of the experimentally measured Young's modulus exhibit a decrease of about two times in comparison with the base Ti substrate where the observed decrease can be attributed to the increase in the unit-cell volumes of $\alpha'$ and $\beta$ structures. Smaller unit-cell volume corresponds to stronger binding forces between the atoms. Thus, the increase in the volumes of the unit cells of $\alpha'$ and $\beta$ can explain the decrease in Young's modulus of Ti–Ta surface alloys formed by electron-beam surface alloying at a beam power of 1750 W. The authors of [19] studied Young's moduli of Ti–Ta ingots formed by melting in an Ar atmosphere. Their results showed that the lowest values were obtained at specimens Ti–30 mass% Ta and Ti–70 mass% Ta. These samples exhibited elastic properties that were the closest to the human bones and are ideal for implant manufacturing. Similarly, the authors of [18] have studied the elastic properties of Ti–Ta alloys prepared by selective laser melting, and the results showed that Young's modulus is in the range of 115 to 89 GPa. The most appropriate specimen for implant manufacturing was established to be Ti–25Ta (wt.%). Similar results were obtained by Zhou et al. [34], where the Ti–25Ta alloy was designed by arc melting and was characterized by Young's modulus of 64 GPa. In our study, the obtained results show even lower values of the measured Young's moduli (i.e., 60.3 ± 5.1 GPa in the case of electron-beam surface alloying with a beam power of 1250 W, and 59.1 ± 16.2 GPa at 1750 W), meaning that it is even closer to that of the human bones (between 9 and 28.4 GPa [35]). Therefore, the obtained surface alloys in the system of Ti–Ta are by electron-beam surface alloying of Ti substrate with Ta film are very promising for implant manufacturing due to the better mechanical compatibility.

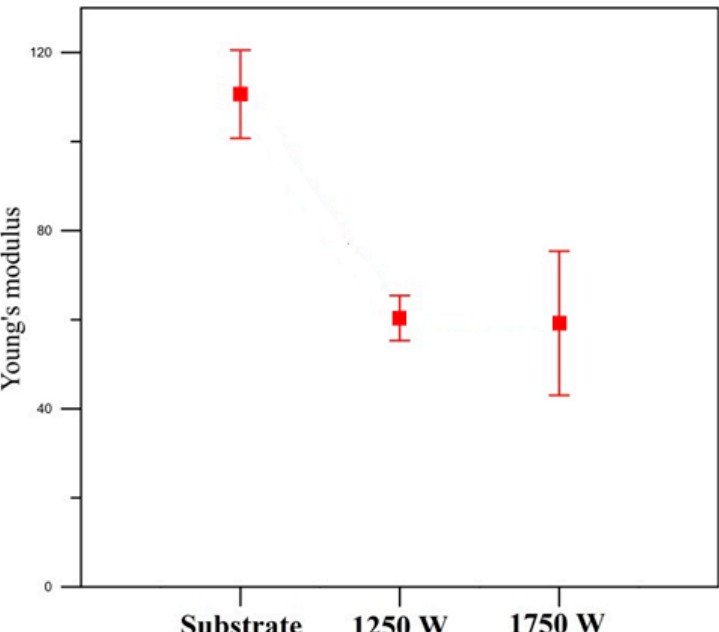

**Figure 6.** Young's modulus of the Ti substrate and the Ti–Ta surface alloys formed by a beam power of 1250 and 1750 W.

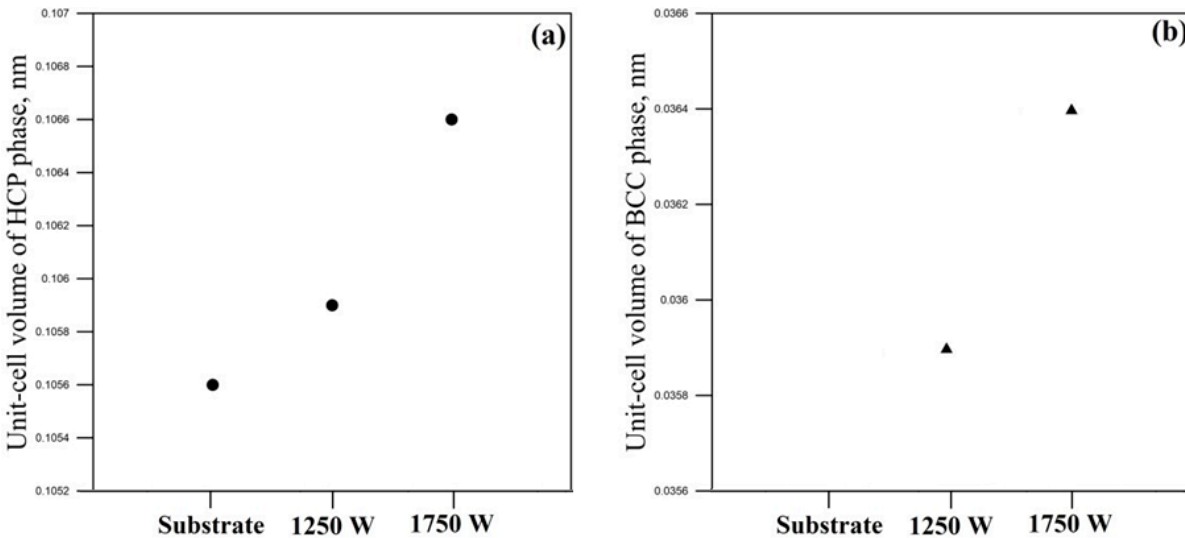

**Figure 7.** Unit-cell volumes of the hcp (**a**) and bcc (**b**) phases of the specimens alloyed with different beam powers.

The coefficient of friction (COF) of the pure Ti substrate and the Ti–Ta surface alloys formed by electron-beam surface alloying with a beam power of 1250 and 1750 W was studied by a micromechanical wear test, and the results of its evolution are presented in Figure 8 and are summarized in Table 2. Further, the average values of the friction coefficient were investigated, and the results reveal that the COF of the Ti substrate was about 0.52; the COF of the Ti–Ta alloy formed by a beam power of 1250 W was about 0.18; the COF of the Ti–Ta alloy formed by a beam power of 1750 W was about 0.36. It is obvious that the higher amount of Ta element within the alloyed zone leads to a decrease in the coefficient of friction. The evolution of the COF from the beginning (i.e., the starting point) to the more advanced moments (i.e., up to the 300th second) is extensively investigated. Considering the Ti substrate, its friction coefficient is about 0.55 at the initial moments of the experiment and stays relatively unchanged to the latter stage of the wear test. The friction coefficient of Ti–Ta surface alloy formed by a beam power of 1250 W is 0.18 at the initial moment of the sliding and remains relatively unchanged during the experiment. The Ti–Ta alloy formed by a beam power of 1750 W is characterized by a COF of 0.17 in the beginning and gradually increases with an increase in the sliding time. At the end of the measurement (i.e., at the 300th second), it reaches a value of 0.48. The results obtained show that the formation of Ti–Ta surface alloy on the top of Ti substrate leads to a decrease in the coefficient of friction, where this effect is more pronounced in the case of the formation of Ti–Ta surface alloys by a beam power of 1250 W. As already mentioned, in this case, the amount of the Ta alloying element is higher in comparison with the electron-beam surface alloyed specimen using a beam power of 1750 W. The lowest friction coefficient in the case of electron-beam surface alloying of Ti substrate with Ta by a beam power of 1250 W could be attributed to the larger amount of the alloying element, where the solid solution strengthening effect is greater [36]. This leads to a lower wear rate and friction coefficient [37–39]. According to the authors of [20], the COF values are significantly lower in the case of Ti–10Ta than the pure Ti material, as well as Ti–5Ta, which confirms the statement that a larger amount of the alloying element corresponds to a lower friction coefficient. The authors of [40] have studied the coefficient of friction Ti–Ta–N coatings deposited on VT6 alloy for medical implants, and the results showed that it is about 0.3. Furthermore, it was found that the COF of the Ti–Ta–N coating is lower than that of the base VT6 alloy. In the study [41], the tribological properties of α+β Ti (Ti6Al4V and Ti6Al7Nb) alloys were studied, and the results showed that the coefficient of friction is in the range from 0.3 to 0.6, depending on the sliding speed and load. Similarly, the authors of [42] have studied the friction coefficient of Ti6Al7Nb alloy, and the COF decreased from 0.58 to 0.42 with an increase in the sliding time and increased with the rise of the applied load.

The results obtained in our study show lower values for COF of the Ti–Ta surface alloys in comparison with other Ti-based α + β materials, which makes them a very promising candidate for implant manufacturing.

**Table 2.** Young's modulus and coefficient of friction of the Ti substrate and Ti–Ta surface alloys formed by a beam power of 1250 and 1750 W.

| Specimen | Young's Modulus, GPa | COF |
|---|---|---|
| Ti substrate | 110 ± 9.8 | 0.52 |
| 1250 W | 60.3 ± 5.1 | 0.18 |
| 1750 W | 59.1 ± 16.2 | 0.36 |

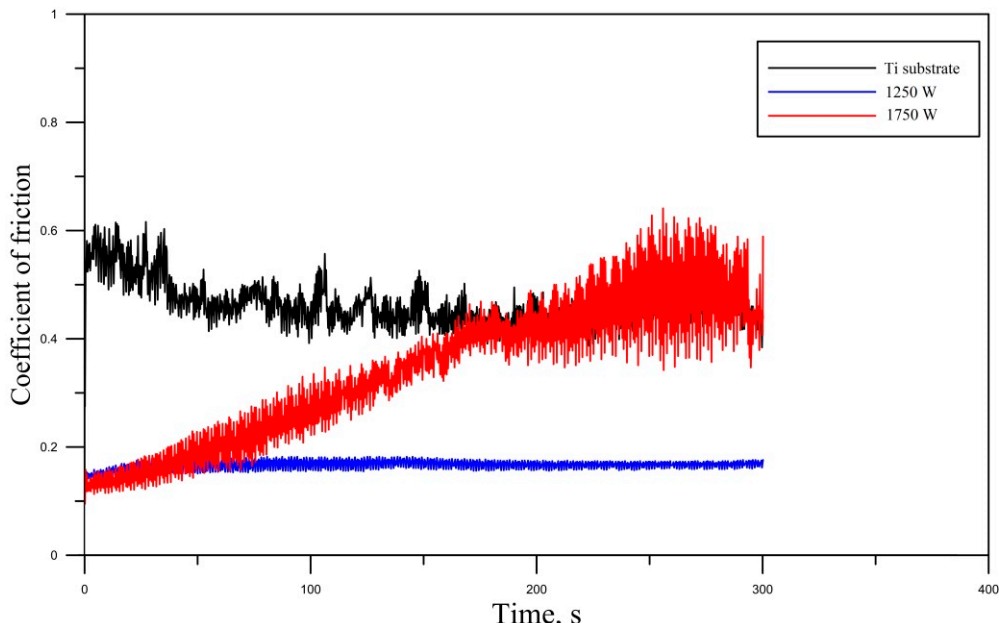

**Figure 8.** Friction coefficients of the Ti substrate and the Ti–Ta surface alloys formed by a beam power of 1250 and 1750 W.

The wear mechanism of the base Ti substrate and Ti–Ta surface alloys formed by a beam power of 1250 and 1750 W was studied, and the worn morphologies of the specimens were analyzed by optical microscopy. Figure 9 shows the worn surfaces of the considered specimens. The base Ti substrate exhibits a large number of furrows and debris, meaning that the wear, in this case, is a mixture of abrasive and adhesive. At the Ti–Ta surface of an alloy formed by a beam power of 1250 W, only furrows with significantly reduced depth are visible, meaning that the wear mechanism, in this case, is abrasive. The worn morphology of the Ti–Ta alloy produced by 1750 W exhibits a large amount of debris, while the furrows are significantly reduced. This means that the wear mechanism is adhesive [43,44]. The results obtained in the present study show that the increase in the Ta content leads to a change in the wear mechanism—from abrasive to adhesive. Similar results were obtained by the authors of [44], where stainless steel was coated with Mo by laser cladding. The increase in the Mo content leads to a change in the wear mechanism from abrasive to adhesive.

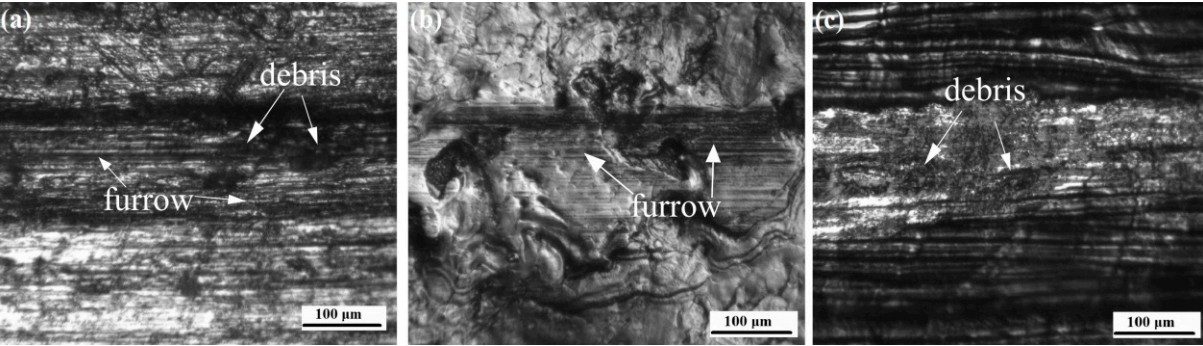

**Figure 9.** Worn surfaces of (**a**) base Ti substrate, (**b**) Ti–Ta alloy formed by a beam power of 1250 W, (**c**) Ti–Ta alloy formed by a beam power of 1750 W.

From a practical point of view, the results obtained in this study for Young's modulus and friction coefficient could have a number of advantages for manufacturing implants and other applications in the field of modern biomedicine. It was demonstrated that Young's modulus is significantly lower in the case of the formed Ti–Ta surface alloy than the base Ti substrate. It reached values of 59.1 GPa, which is about two times lower in comparison with the Ti material. As mentioned in the introduction, excellent biomedical implants require low Young's moduli, of which values should be much closer to that of human bones. In the present particular case, Young's moduli of the Ti–Ta surface alloys are significantly lower (about two times) than the pure Ti substrate. The authors of [19] have summarized the elastic properties of some typical implant materials. The results from [19] showed that the Ti–Ta alloys exhibit the lowest Young's moduli, where the mentioned values are 69 GPa in the case of Ti–30 mass% Ta, and 67 GPa for Ti–70 mass% Ta. These alloys are expected to have the potential to be a new candidate for biomedical applications and implant manufacturing. In our study, the obtained results show even lower values of the measured Young's moduli (i.e., $60.3 \pm 5.1$ GPa in the case of electron-beam surface alloying with a beam power of 1250 W, and $59.1 \pm 16.2$ GPa at 1750 W), meaning that it is even closer to that of the human bones (between 9 and 28.4 GPa [35]). This significant decrease in the values of the elastic modulus is an important desirable effect for the implementation of the Ti–Ta surface alloys formed by selective electron-beam alloying. On the other hand, it is known that the implants are exposed to significant friction when are inserted into the human body. This causes a separation of metallic ions and particles. This could result in adverse reactions and implant failure. Therefore, the problems related to the reduction in the coefficient of friction and improvement in the friction properties of the implant materials are of major importance. The results obtained in the present study for the COF show that it is significantly improved in the case of a Ti–Ta surface alloy formed by electron-beam surface, alloying by a beam power of 1250 W. As already mentioned, it decreases about two times in comparison with the base Ti substrate. These results are very important from a practical point of view, and they are expected to open new potential applications of these materials.

The results obtained in the present study confirm that the Ti–Ta alloys are very promising for manufacturing implants and can be characterized as a material with high mechanical compatibility. However, it was found that the highest amount of the alloying element (i.e., Ta) is about 19 wt.%. Up to now, there are no data for electron-beam manufactured surface Ti–Ta alloys on Ti substrate with higher Ta content. Therefore, the questions related to the manufacturing of such an implant material by electron-beam surface alloying, where the Ta concentration is higher than 19 wt.%, as well as the influence of this higher content, is still open.

## 4. Conclusions

In this study, we present results of Young's modulus and coefficient of friction of Ti–Ta alloys formed by electron-beam surface alloying of Ti substrate with Ta film by a scanning electron beam. During the alloying process, the beam power was varied from 750 to 1750 W. It was found that at 750 W, the Ta film remained undissolved on the top of the Ti, and no alloyed zone has been observed. By an increase in the beam power to 1250 and 1750 W, a distinguished alloyed zone is formed, where it is much thicker in the case of 1750 W. It was demonstrated that the amount of the Ta element within the alloyed zone obtained at 1250 W is larger and some amount of undissolved particles exist. The phase composition of the obtained surface alloys is in the form of double-phase α'and β. It was found that in the case of alloying by a beam power of 1250 W, the amount of the beta phase is larger than that of the surface alloy formed by 1750 W. The electron-beam surface alloying process leads to a decrease in Young's modulus of about two times in comparison with the base Ti substrate. In the case of alloying by a beam power of 1250 W, the observed drop is attributed to the larger amount of the β phase, while at 1750 W, it is due to the weaker binding forces between the atoms. The results obtained for the coefficient of friction show that the formation of Ti–Ta surface alloy on the top of Ti substrate leads to a decrease in the coefficient of friction, where the effect is more pronounced in the case of the formation of Ti–Ta surface alloys by a beam power of 1250 W.

**Author Contributions:** Conceptualization, S.V. and P.P.; methodology, S.V., D.D., N.I., R.B. and M.O.; formal analysis, S.V., D.D., N.I., R.B., and M.O.; investigation, S.V., D.D., N.I., R.B., and M.O.; writing—original draft preparation, S.V.; writing—review and editing, S.V.; supervision, P.P. All authors have read and agreed to the published version of the manuscript.

**Funding:** This research was funded by Bulgarian National Science Fund, grant number KP-06-M37/5.

**Institutional Review Board Statement:** Not applicable.

**Informed Consent Statement:** Not applicable.

**Data Availability Statement:** Not applicable.

**Acknowledgments:** In memoriam of our great teacher and scientific supervisor, Peter Petrov.

**Conflicts of Interest:** The authors declare no conflict of interest.

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
