# Peer review of "Influence of Beam Power on Young’s Modulus and Friction Coefficient of Ti–Ta Alloys Formed by Electron-Beam Surface Alloying"

_metals, doi:10.3390/met11081246_

Round 1

Reviewer 1 Report

In this study, the authors presented results of Young’s modulus and coefficient of friction of Ti-Ta alloys formed by electron-beam surface alloying of Ti substrate with Ta film by a scanning electron beam. The result is interesting and they are expected to open new potential applications for Ti alloy. A few questions and suggestions that have to be answered before publication as follows.

  1. Figure 6 does not make the dot-line diagram of the three data standardized. A scatter chart should be used.
  2. The abscissas of Fig. 6 and Fig. 7 are not uniform.
  3. Is there a standard for micromechanical friction testing? The micromechanical wear test time is too short. The friction coefficient is unstable.
  4. Is there any data on wear volume or mass loss to reflect the relative wear resistance?
  5. What is the wear mechanism of the Ta-Ti surface? Is it possible to add the corresponding figures of worn surface? For example, Effect of Molybdenum on the Microstructures and Properties of Stainless Steel Coatings by Laser Cladding. Applied Sciences, 2017,7(10): 1065. Microstructure and property of laser clad Fe-based composite layer containing Nb and B4C powders. Journal of Alloys and Compounds, 2019,802: 373-384.

Author Response

Dear reviewer,

Thank you for the valuable comments and suggestions that helped us improve the quality of our manuscript! Attached you can find our responses to your comments and questions.

Best regards,

Stefan Valkov

Reviewer 2 Report

I have no major comments on the submitted paper 

Author Response

The authors would like to thank the reviewer for the comment.

Reviewer 3 Report

I read this work and found it is interesting based on detailed invesitigations, and therefore deserve the publication in this journal. With regard to the element analysis, it is suggested that advanced tool of synchrotron radiation source, and elemental effect can be given more literature, please refer to the paper "The imaging of failure in structural materials by synchrotron radiation X-ray micro-tomography" and "Porosity, element loss and strength model on softening behavior of hybrid laser arc welded Al-Zn-Mg-Cu alloy with synchrotron radiation analysis" into your background. Besides, I also suggested the authors to give more detailed parameter of analyzed machine such as SEM\XRD\EDS etc. Base on the quality of this paper, I recommend the correction after minor revision. 

Author Response

The authors would like to thank the reviewer for the valuable comments and suggestions. Corresponding information and the suggested references were added to the introduction of the manuscript. Also, more information about SEM/EDX and XRD measurements and machines was added to "Materials and methods".

Once again we would like to thank you for the valuable comments and suggestions that helped us improve the quality of our manuscript.

Reviewer 4 Report

  1. The authors of the paper should specify, based on previous studies as well as the literature review, what specific values of Young's modulus and COF they expect.
  2. Why do they include results with beam power 750 W which practically did not give positive results? The presented results indicate a research report rather than a detailed scientific analysis of the research result
  3. No significant evaluation of how the results were achieved compared to the properties of Ti-Ta alloys obtained with other technologies. 
    5. in the opinion of the authors, should there be further research in this field?

Author Response

Dear reviewer,

The authors would like to thank the reviewer for the valuable comments and suggestions. Attached you can find our responses to your comments and questions.
